# Magnesium: An overlooked signalling ion in plant physiology and circadian regulation

Charlotte Wathar and Nathalie Verbruggen

Laboratory of Plant Physiology and Molecular Genetics, Université libre de Bruxelles (ULB), Belgium

cicardian clock; energy; metabolism; nutrient homeostasis; Tor.

**Corresponding author:**
Nathalie Verbruggen;
Email: Nathalie.Verbruggen@ulb.be

**Associate Editor:**
Prof. Ingo Dreyer

## Abstract

Magnesium ($Mg^{2+}$) is essential for plant growth and metabolism, acting as a cofactor in numerous enzymatic and structural processes. This review outlines the main physiological and biochemical functions of $Mg^{2+}$ and summarizes current knowledge on its transport and homeostatic regulation. We examine how $Mg^{2+}$ homeostasis intersects with broader signalling networks and metabolic pathways, including its crosstalk with other mineral nutrients, where antagonistic and synergistic interactions influence nutrient acquisition, allocation and stress responses. Emerging evidence further suggests that, beyond its classical roles, $Mg^{2+}$ may function as a regulatory ion with signalling properties reminiscent of secondary messengers in animal systems. Finally, we highlight recent findings linking $Mg^{2+}$ dynamics to circadian regulation, suggesting reciprocal interactions between temporal control mechanisms and nutrient fluxes. These insights underscore the central importance of $Mg^{2+}$ in plant biology and identify key gaps in understanding its regulatory and integrative roles.

## 1. Introduction

Magnesium is a macronutrient absorbed by plants in its ionic form $Mg^{2+}$, playing fundamental physiological roles in plants and all other living organisms. It is the eighth most abundant element on Earth and the most abundant divalent cation in all living cells (Culkin & Cox, 1966; Fleischer, 1954; Maguire & Cowan, 2002; Senbayram et al., 2015). Today, $Mg^{2+}$ deficiency in soils is an increasingly worrying problem, which limits crop yields and affects human nutrition, as plants are our main source of $Mg^{2+}$. For many years, this element has been under-investigated, therefore earning the name of the 'forgotten element' (Broadley & White, 2010; Cakmak & Yazici, 2010; Fan et al., 2008; Hermans et al., 2013; Shaul, 2002). Up to 30% of adults in Europe and North America fail to meet the Estimated Average Requirement for $Mg^{2+}$ (Rosanoff, 2013). More recent assessments confirmed chronic low dietary $Mg^{2+}$ is widespread and linked to public health risks (Adomako & Yu, 2024; Wu et al., 2025). Symptoms of hypomagnesemia (serum $Mg^{2+} < 0.75$ mmol/L) include extreme fatigue, muscle dysfunction, more risk of cardiovascular disease, arrhythmia, cardiac death, insulin resistance and hypertension (Rosanoff, 2013; Al Alawi et al., 2018; Wu et al., 2025).

In this review, we briefly outline the main physiological and biochemical roles of $Mg^{2+}$ and we provide an overview of the key components involved in $Mg^{2+}$ transport and homeostasis. In addition, we examine how $Mg^{2+}$ homeostasis intersects with broader cellular signalling networks and metabolic processes. Of particular interest are the interactions between $Mg^{2+}$ homeostasis and circadian clocks, raising important questions about how temporal control mechanisms influence nutrient fluxes and how $Mg^{2+}$ dynamics, in turn, feed back into the circadian clock. By coordinating key physiological processes that influence crop yield and resource efficiency, the circadian clock holds great potential for 'chronoculture', an approach to agriculture that harnesses biological timing to optimize practices, such as applying nutrients at the time of day when they are most effective (Gerhardt & Mehta, 2025; Ogasawara et al., 2025; Steed et al., 2021).

## 2. Main physiological and biochemical roles of $Mg^{2+}$ and symptoms of deficiency

The physiological importance of $Mg^{2+}$ has been most extensively characterized in the context of photosynthesis. As the central atom of chlorophyll, $Mg^{2+}$ directly participates in light capture.

It also stabilizes pigment-protein complexes in Photosystems I and II and contributes to the maintenance of thylakoid membrane architecture. Because thylakoid membranes carry a negative surface charge, $Mg^{2+}$ plays a crucial role in counteracting these charges, promoting the appression of adjacent membranes and ultimately driving grana stacking. When $Mg^{2+}$ becomes limiting, electrostatic repulsion increases and grana becomes unstacked or disorganized. As a result, chlorophyll content is reduced and interveinal chlorosis develops, predominantly in older mature leaves, accompanied by a decrease in photosynthetic rate (Cakmak & Yazici, 2010; Levitt, 1954; Marschner & Marschner, 2012; Pandey, 2018). Notably, however, chlorosis is a late symptom of $Mg^{2+}$ shortage (Hermans et al., 2004; Ogura et al., 2020)

Beyond photosynthesis, $Mg^{2+}$ is essential for ribosome biogenesis and translation. It stabilizes ribosomal subunits and facilitates the binding of messenger RNA (mRNA) and transfer RNA (tRNA) (Klein et al., 2004; Kobayashi & Tanoi, 2015; Sperrazza & Spremulli, 1983). When $Mg^{2+}$ is deficient, these processes are impaired, leading to reduced protein synthesis (de Melo et al., 2021; Feeney et al., 2016). This effect is especially pronounced in chloroplasts, which house more than 25% of total cellular proteins (Marschner & Marschner, 2012; Peng et al., 2015; Chen et al., 2018).

In addition to its function in translation, $Mg^{2+}$ serves as an essential cofactor and allosteric modulator for more than 300 enzymes involved in respiration, glycolysis, nucleic acid metabolism, chlorophyll biosynthesis and photosynthetic carbon fixation (De Bang et al., 2021; Hermans et al., 2013). Key examples include ribulose-1,5-bisphosphate carboxylase/oxygenase (Rubisco) and phosphoenolpyruvate (PEP) carboxylase, underscoring the central role of $Mg^{2+}$ in carbon assimilation (Andreo et al., 1987; Lorimer et al., 1976; Walker & Weinstein, 1994; Willows, 2003). $Mg^{2+}$ is also indispensable for energy metabolism as the binding partner of adenosine triphosphate (ATP). Between 50% and 90% of cytosolic $Mg^{2+}$ is complexed with ATP, and $Mg^{2+}$ scarcity impairs Mg-ATP formation, thereby compromising energy-dependent reactions across metabolic pathways (Maguire & Cowan, 2002; Marschner & Marschner, 2012; Shaul, 2002). Moreover, ATP synthesis itself is highly dependent on $Mg^{2+}$, since ATP synthase requires $Mg^{2+}$ as a cofactor to catalyse the phosphorylation of ADP (Gout et al., 2014; Lin & Nobel, 1971).

A key outcome of $Mg^{2+}$ deficiency is impaired phloem loading. Reduced activity of $H^+$-ATPases in phloem companion cells compromises the proton motive force and restricts sucrose export from source to sink tissues in species relying on apoplastic loading of photoassimilates (Bush, 1989; Cakmak & Yazici, 2010; Vaughn et al., 2002). Experimentally, $Mg^{2+}$ deficiency has been shown to induce carbohydrate accumulation in source leaves, increase the shoot-to-root dry weight ratio (Cakmak et al., 1994; Cakmak & Kirkby, 2008; Hermans et al., 2004), repress photosynthetic gene expression and further reduce the activities of $CO_2$-assimilating and photosynthetic enzymes (Cakmak & Kirkby, 2008; Chen et al., 2018).

At the structural level, prolonged $Mg^{2+}$ deficiency leads to swelling and disorganization of chloroplasts, as well as damage to the photosynthetic electron transport chain (Hermans et al., 2004; Laing et al., 2000). These alterations promote overproduction of reactive oxygen species (ROS) (Cakmak, 1994; Guo et al., 2016), which in turn exacerbate disturbances in source-sink dynamics, growth and biomass allocation (Hermans et al., 2004, 2013; Kobayashi & Tanoi, 2015; Peng et al., 2015; Koch et al., 2020). High-light environments amplify these detrimental effects by accelerating photooxidative stress and further enhancing ROS

production (Cakmak & Kirkby, 2008; Kumar Tewari et al., 2006; Marschner & Marschner, 2012; Ye et al., 2024).

The distribution of $Mg^{2+}$ within leaves also reflects its nutritional status. Depending on $Mg^{2+}$ availability, up to 35% of total leaf $Mg^{2+}$ is found in chloroplasts, with up to 25% associated to chlorophyll; this percentage is increased by $Mg^{2+}$ deficiency or low-light conditions (Broadley & White, 2010; Cakmak & Yazici, 2010; Dorenstouter et al., 1985; Scott & Robson, 1990). The remainder is distributed among the vacuole (the main storage site), the cell wall and the water-soluble cytoplasmic pool (Leigh & Wyn Jones, 1986; Maguire & Cowan, 2002; Marschner & Marschner, 2012). The vacuole's strong $Mg^{2+}$ retention capacity limits both deficiency and toxicity (Shaul, 2002; Stelzer et al., 1990).

Finally, accumulating evidence indicates that $Mg^{2+}$ supplementation can enhance plant resistance to both abiotic stresses, such as high light, cold, drought and heat, and biotic stresses, including fungal and bacterial infections (Shao et al., 2021; Gupta et al., 2022; Li et al., 2023). For example, $Mg^{2+}$ oxide (MgO) application in tomato induces immunity against Fusarium wilt by activating the jasmonic acid signalling pathway (Fujikawa et al., 2021), and overall $Mg^{2+}$ fertilization can improve crop yield (Wang, Hassan, et al., 2020). Consequently, $Mg^{2+}$ deficiency not only disrupts photosynthesis and carbohydrate allocation but also alters broader nutrient homeostasis (see Section 3), plant stress responses and immunity, thereby reinforcing a feedback loop of physiological decline. Understanding these interdependencies is therefore critical for optimizing fertilization strategies, particularly in soils with variable pH, texture and ionic composition.

## 3. Magnesium availability in soils and interactions with other nutrients

In soils, adequate $Mg^{2+}$ concentrations range between 0.12 and 8.5 mM to sustain optimal plant growth (Karley & White, 2009; Marschner & Marschner, 2012). Levels outside this range can cause deficiency or toxicity, both of which impair photosynthesis, carbohydrate partitioning and biomass accumulation (Farhat et al., 2016; Guo et al., 2016). Therefore, understanding the geochemical, physiological and competitive factors regulating $Mg^{2+}$ availability is critical to improve yield in agricultural systems. Importantly, soil and plant $Mg^{2+}$ concentrations are shaped by numerous chemical, mineralogical and environmental factors (Senbayram et al., 2015).

A major determinant of soil $Mg^{2+}$ concentration is the mineral composition and degree of weathering. Primary silicates like olivine and augite contain more $Mg^{2+}$ than highly weathered minerals such as muscovite (Guo, 2017; Maguire & Cowan, 2002; Mayland & Wilkinson, 1989). In addition, due to its large hydrated ionic radius and low binding affinity for negatively charged soil colloids, $Mg^{2+}$ is highly mobile and easily leached, particularly in sandy or acidic soils (Guo, 2017; Maguire & Cowan, 2002). Acidic soils, which represent nearly 70% of global arable land, often have low cation exchange capacity, limiting $Mg^{2+}$ retention. This problem is exacerbated under high rainfall or irrigation, further increasing the risk of $Mg^{2+}$ deficiency (Aitken et al., 1999; Mengel et al., 2001; Senbayram et al., 2015).

However, $Mg^{2+}$ availability is not determined solely by its absolute concentration but also by its interactions with other ions in the rhizosphere and plant tissues. Such interactions can influence uptake, transport or utilization through precipitation, complexation or competition for binding and transport sites (Epstein & Bloom, 1972; Fageria, 2001; Marschner & Marschner,

2012). For instance, $Mg^{2+}$ forms insoluble $Mg^{2+}$ carbonate ($MgCO_3$) in alkaline or calcareous soils (Broadley & White, 2010) and often competes with cations of similar properties, including $Ca^{2+}$, $K^+$, $Na^+$ and $NH_4^+$, to maintain electrochemical and osmotic balance (Gransee & Führs, 2013; Lasa et al., 2000; Peuke et al., 2002). This mutual antagonism is especially relevant in the context of modern agriculture, where unbalanced use of nitrogen–phosphorus–potassium (NPK) fertilizers contributes to progressive $Mg^{2+}$ depletion in soils (Cakmak & Yazici, 2010; Hermans et al., 2013; Kobayashi & Tanoi, 2015).

$Mg^{2+}$ interactions with other ions can also be synergistic, enhancing physiological functions or antagonistic, where excess of one ion inhibits the uptake of others. The effects depend on soil properties, nutrient ratios, plant species, plant tissues, developmental stage and environmental conditions (Chaudhry et al., 2021; Hermans et al., 2013). At low to moderate concentrations, $Mg^{2+}$ uptake shows synergistic or neutral interactions with $Ca^{2+}$ and $K^+$ but becomes antagonistic when $Ca^{2+}$ or $K^+$ are present in excess. Under $Mg^{2+}$ limitation, antagonism is usually observed, although findings for $K^+$ remain inconsistent (Heenan & Campbell, 1981; Fageria, 1983; Lasa et al., 2000; Hermans et al., 2004; Ding et al., 2006; Tang & Luan, 2017; Rhodes et al., 2018; Xie et al., 2021; Garcia et al., 2022). Interactions with micronutrients such as $Mn^{2+}$, $Zn^{2+}$, $Fe^{2+}$ and $Cu^{2+}$ are context-dependent: high concentrations of these metals can inhibit $Mg^{2+}$ uptake, whereas $Mg^{2+}$ deficiency may induce their accumulation or have little effect (Heenan & Campbell, 1981; Kumar et al., 1981; Agarwala et al., 1988; Le Bot et al., 1990; Lu et al., 2021; Sadeghi et al., 2021; Xu et al., 2024). $Mg^{2+}$ generally supports P acquisition, while deficiency reduces or has little impact on tissue P content (Fageria, 1983; Skinner & Matthews, 1990; Ogura et al., 2020; Weih et al., 2021). The form of N also affects $Mg^{2+}$ uptake: $NH_4^+$ tends to inhibit it, whereas $NO_3^-$ promotes acquisition. Conversely, $Mg^{2+}$ fertilization can enhance $NO_3^-$ uptake by upregulating nitrate transporter genes such as *NRT2.1* and *NRT2.2* (Mayland & Wilkinson, 1989; Lasa et al., 2000; Peng et al., 2020; Tian et al., 2023). Finally, $Mg^{2+}$ often mitigates toxic ions such as $Al^{3+}$, $Cd^{2+}$, $Na^+$ and $Li^+$, exerting protective effects by competing for transporters, stabilizing membranes, promoting organic acid exudation and maintaining cytoplasmic pH or metal homeostasis. For example, the overexpression of $Mg^{2+}$ transporters genes (*AtMGT1* and *OsMGT1*) increases cytosolic $Mg^{2+}$ and enhances tolerance to $Al^{3+}$ stress (Deng et al., 2006; de Wit et al., 2010; Chou et al., 2011; Hermans et al., 2011; Rengel et al., 2015; Chen et al., 2017; Lyu et al., 2023; Garcia-Daga et al., 2025).

Taken together, the multiple cellular roles of $Mg^{2+}$ emphasize the need for strict control of its concentration and distribution within plant tissues and organelles. These functional requirements have driven the evolution of specialized transport systems and regulatory networks that ensure $Mg^{2+}$ homeostasis.

# 4. Magnesium transport systems in plants

While $Mg^{2+}$ concentration in plants is estimated at 25 mM, most of it is tightly bound to nucleotides, ribosomes, chlorophyll or stored in vacuoles, making only a small fraction (0.2–5 mM) available for metabolism. This free $Mg^{2+}$ pool is dynamically regulated by $Mg^{2+}$ transporters and the binding of $Mg^{2+}$ to nucleotides involved in cellular reactions (Kleczkowski & Igamberdiev, 2021, 2023).

$Mg^{2+}$ transport in plants involves coordinated uptake, long-distance translocation and intracellular partitioning, ensuring proper allocation across tissues and organelles. The core $Mg^{2+}$ transport system in higher plants is formed by the Mitochondrial RNA Splicing 2 (MRS2)/$Mg^{2+}$ transporter (MGT) family, the $Mg^{2+}$/H+ exchanger (MHX) and members of the Magnesium release (MGR)/Cyclin M (CNNM)/Cation Outward Rectifier C (CorC) family, which are more recently identified key players of $Mg^{2+}$ transport. Additional systems like cyclic nucleotide–gated ion channels (CNGCs) contribute in specific contexts.

## 4.1 Uptake from soil and transport in the root

$Mg^{2+}$ uptake begins at the root–soil interface, where $Mg^{2+}$ is absorbed predominantly as a free ion. Transport across the plasma membrane of root epidermal cells, particularly root hairs, is mediated mainly by members of the MRS2/MGT family. This family comprises three subfamilies, the CorA-like (CorA is the name of the family in prokaryotes because of mutants exhibiting $Co^{2+}$ resistance), the Nonimprinted in Prader-Willi/Angelman syndrome (NIPA), and the membrane $Mg^{2+}$ transporters (MMgT), based on conserved-motifs distribution. The major group of $Mg^{2+}$ transporters in plants is characterized by CorA-like domains and is named MRS2/MGT, while the physiological roles of NIPA and MMgT homologs in plants remain to be experimentally validated (Anwar et al., 2025; Mohamadi et al., 2023). MRS2/CorA are homopentameric channels with two transmembrane (TM) helices in each monomer. They are characterized by a conserved Gly-X-Asn (GxN) motif, of which the X represents hydrophobic amino acids Met, Val or Ile, that forms part of the ion-conducting pore and is critical for $Mg^{2+}$ selectivity and transport activity (Moomaw & Maguire, 2008; Ishijima et al., 2021; Franken et al., 2022; Li et al., 2025). MGT/MRS2 transporters are functionally conserved across plant species, and members have been identified among others in Arabidopsis, rice, wheat, maize, tomato and cucumber, highlighting both conservation and diversification of roles in $Mg^{2+}$ homeostasis. These transporters exhibit varying affinities and selectivities, enabling plants to efficiently absorb $Mg^{2+}$ across a wide range of external concentrations.

In Arabidopsis, MRS2-4/MGT6, which localizes to the plasma membrane of root hairs, functions as a major high-affinity $Mg^{2+}$ uptake system under $Mg^{2+}$ limitation (Mao et al., 2014; Tang et al., 2022; Yan et al., 2018). Loss-of-function *mgt6* mutants exhibit strong growth defects and chlorosis under low $Mg^{2+}$, while overexpression enhances $Mg^{2+}$ uptake capacity, confirming its central role in root $Mg^{2+}$ acquisition. MRS2-7/MGT7, localized in the endomembrane system of root cells, is proposed to contribute to intracellular $Mg^{2+}$ partitioning and homeostasis (Gebert et al., 2010). Combined loss of *MGT6* and *MGT7* led to exacerbated phenotype compared with single mutants under both $Mg^{2+}$-deficient and $Mg^{2+}$-excess conditions (Yan et al., 2018). In rice, the orthologue of AtMGT6, OsMGT1, has been demonstrated to mediate root $Mg^{2+}$ uptake: knockout plants display decreased root $Mg^{2+}$ uptake, lower tissue $Mg^{2+}$ content and reduced biomass under $Mg^{2+}$ deficiency, whereas overexpression increases $Mg^{2+}$ under low $Mg^{2+}$ supply (Chen et al., 2012; Zhang, Peng, et al., 2019). Orthologous genes have also been identified in tomato (*SlMGTs*; Regon et al., 2019; Liu et al., 2023), though their functional roles remain less well characterized.

A novel functional gene, *LOC_Os03g04360*, annotated as a putative inorganic phosphate transporter belonging to the *OsPHT1* gene family, controlling $Mg^{2+}$ uptake and translocation in rice was identified using QTL analysis (Zhi et al., 2023). Overexpression of *LOC_Os03g04360* could significantly increase

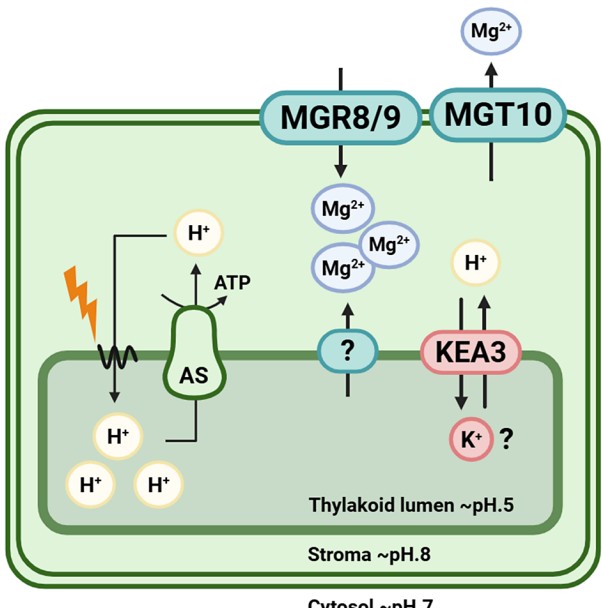

**Figure 1.** Schematic overview of chloroplastic $Mg^{2+}$ flux under illumination in Arabidopsis.

Upon illumination, photosynthetic electron transport drives proton pumping from the stroma into the thylakoid lumen, generating a steep trans-membrane proton gradient (lumen ~pH 5; stroma ~pH 8). This proton motive force powers ATP synthase (AS), which restores $H^+$ to the stroma while producing ATP. Lightning symbol indicates light-driven reactions of the photosynthetic electron transport chain. The resulting stromal alkalinization triggers $Mg^{2+}$ release from the thylakoid lumen into the stroma, thereby increasing stromal $Mg^{2+}$ concentration. At the chloroplast inner envelope, $Mg^{2+}$ import from the cytosol into the stroma is mediated by transporter proteins MGR8 and MGR9, supporting $Mg^{2+}$ requirements for ATP stabilization, enzyme activation, chlorophyll-binding protein function and other chloroplastic-related functions. The AtMGT10/OsMGT3 channel is likely responsible for $Mg^{2+}$ efflux or exchange across the inner envelope, maintaining charge balance during light-dependent $H^+$ movements. After a sudden light decrease, KEA3-driven $H^+$ export from the lumen to the stroma in exchange for another cation, most likely $K^+$, is required for the prompt relaxation of non-photochemical quenching. Since the substrate of KEA3 has not been demonstrated in plants, transport of $Mg^{2+}$ via the antiporter KEA3 cannot be excluded (Uflewski et al., 2021) *(figure created with BioRender.com)*.

the $Mg^{2+}$ concentration in rice seedlings, especially under the condition of low $Mg^{2+}$ supply, but it decreased Mg content ratio of shoot to root.

Once absorbed, $Mg^{2+}$ is transported to different tissues in the roots through the apoplastic and symplastic pathways.

### 4.2. Loading into and transport via the Xylem and Phloem

After entering root cells, $Mg^{2+}$ must be loaded into the xylem for delivery to aerial tissues. In Arabidopsis, four plasma membrane-localized transporters, MGR4-MGR7, are essential for root-to-shoot $Mg^{2+}$ allocation by mediating its release into the xylem (Meng, Zhang, Tang, et al., 2022).

Additional evidence implicates the cyclic nucleotide-gated channel 10 (CNGC10). CNGC proteins are non-specific ligand-gated $Ca^{2+}$-permeable channels (Talke et al., 2003). Suppression of *AtCNGC10* resulted in altered shoot ion profiles, with decreased $Ca^{2+}$ and $Mg^{2+}$ but elevated $K^+$, suggesting that this channel contributes to long-distance cation transport, potentially through roles in xylem loading/retrieval and/or phloem loading (Guo et al., 2010). *AtCNGC10* is more highly expressed in roots than in leaves (Christopher et al., 2007), the channel is in the plasma membrane, but its precise tissue localization and specific transport function remain unresolved today.

Beyond xylem transport, the phloem plays a central role in redistributing $Mg^{2+}$ between source and sink tissues, particularly during deficiency or senescence. Because $Mg^{2+}$ is phloem-mobile, deficiency symptoms characteristically develop first in mature fully expanded leaves, rather than in young leaves (De Bang et al., 2021;

Hermans et al., 2004; Hermans & Verbruggen, 2005; Ogura et al., 2020).

### 4.3. Subcellular transport

$Mg^{2+}$ is vital at the subcellular level, and its transport has been especially studied in chloroplasts where it is the central atom of chlorophyll and participates in photosynthetic enzyme function and thylakoid stacking. Internal $Mg^{2+}$ concentration can be measured by using an $Mg^{2+}$-sensitive fluorescent indicator, mag-fura-2. Upon illumination, release of $Mg^{2+}$ from thylakoid membranes has been observed in intact chloroplasts, and $Mg^{2+}$ concentration typically increases from 0.5 mM to 2.0 mM in the stroma (Ishijima et al., 2003). Stromal ion homeostasis depends on the activity of both thylakoid and envelope ion channels. While the identity of the $Mg^{2+}$ transport system across the thylakoid membrane is unknown, transport across chloroplast inner envelope is mediated by AtMRS2-11/AtMGT10 in Arabidopsis (with OsMGT3 as its orthologue in rice) and the MGR family members MGR8 and MGR9 (ACDP/CNNM/CorC-related) (Figure 1). These three transporters have different roles, as evidenced by distinct phenotypes of their respective mutants (Dukic et al., 2023). For instance, thylakoid stacking is disrupted in *mgt10* mutants but remains largely unaffected in *mgr8* or *mgr9*. Conversely, grana size is reduced in the *mgr8 mgr9* double mutant (Dukic et al., 2023; Zhang et al., 2022) but increased in *mgt10* plastids (Dukic et al., 2023; Sun et al., 2017). In Arabidopsis, AtMRS2-11/MGT10 has been shown to mediate $Mg^{2+}$ export from chloroplasts using dye-based $Mg^{2+}$ imaging, whereas MGR8 and MGR9 are responsible for $Mg^{2+}$ import into the stroma (Ishijima et al., 2021; Kunz

et al., 2024). In addition to these envelope transporters, the thylakoid cation/H⁺ antiporter KEA3 may also contribute to $Mg^{2+}$ movements within the chloroplast. KEA3 exports protons from the lumen in exchange for another cation, most likely $K^+$, yet transport of $Mg^{2+}$ cannot be excluded (Uflewski et al., 2021).

Vacuolar sequestration of $Mg^{2+}$ is another important aspect of intracellular $Mg^{2+}$ partitioning. The MHX transporter, a vacuolar metal/H⁺ exchanger, contributes mainly to detoxification and buffering of excess $Mg^{2+}$, along with other divalent cations such as $Zn^{2+}$ and $Cd^{2+}$ (Shaul, 2002). Conn et al. (2011) suggested that MGT2/MRS2-1 and MGT3/MRS2-5 might contribute to vacuolar accumulation of $Mg^{2+}$ in the mesophyll cells, especially under serpentine conditions (high $Mg^{2+}/Ca^{2+}$ ratio in the soil) (Conn et al., 2011). However, subsequent work rather showed that MGT2 functions in $Mg^{2+}$ efflux from the vacuole (as detailed below), whereas the tonoplast MGR1 mediates $Mg^{2+}$ sequestration into the vacuole (Tang et al., 2022).

The remobilization of $Mg^{2+}$ from the vacuole to the cytoplasm is another key step in $Mg^{2+}$ homeostasis. This process is mediated by MRS2-10/MGT1 and MRS2-1/MGT2 in Arabidopsis, two redundant vacuolar transporters (only the double mutant displays a phenotype) (Tang et al., 2022). Note that AtMGT1 was initially mislocalized in the plasma membrane (Li et al., 2001). $Mg^{2+}$ efflux via MGT1 and MGT2 constitutes a rate-limiting step in $Mg^{2+}$ remobilization from old leaves to young tissues such as seeds (Tang et al., 2022).

## 5. Magnesium homeostasis signalling pathways

Magnesium homeostasis ensures that plants maintain an optimal internal $Mg^{2+}$ concentration, supporting metabolic processes while preventing excessive accumulation that could become toxic. The main mechanisms contributing to $Mg^{2+}$ homeostasis include regulated uptake by roots, controlled long-distance transport and tissue partitioning, dynamic storage and remobilization from intracellular compartments and signalling pathways that adjust transporter activity according to both internal and external $Mg^{2+}$ status.

Regulation of $Mg^{2+}$ transporters operates at multiple levels, including transcriptional control (Franken et al., 2022). However, unlike other macro-nutrients, $Mg^{2+}$ deficiency generally induces only limited transcriptional regulation of the $Mg^{2+}$ transporter genes involved in root uptake (Gebert et al., 2010; Hermans, Vuylsteke, Coppens, Craciun, et al., 2010; Ogura et al., 2020). While Ogura et al. (2018) showed that $Mg^{2+}$ uptake system was up-regulated in roots within 1 h in response to low $Mg^{2+}$, this induction was not seen in *mgt6* or *mgt7* mutants, and the expression of *AtMRS2-4*/*AtMGT6* and *AtMRS2-7*/*AtMGT7* was not responsive to this condition (Ogura et al., 2018). In rice, *OsMGT1* was only induced in shoots but not in roots under $Mg^{2+}$ deficiency (Zhang, Peng, et al., 2019). However, differences between species or experimental conditions may occur, as illustrated in tomato, where $Mg^{2+}$ deficiency led to a clear transcriptional induction of several $Mg^{2+}$ transporter genes (*SlMGT1*, *SlMGT6*, *SlMGT7*, and *SlMGT10*) in roots (Ishfaq et al., 2021). Overall, this suggests that $Mg^{2+}$ uptake is primarily controlled by post-transcriptional or post-translational mechanisms or by changes in transporter activity and localization rather than gene induction.

A key regulation is by $Mg^{2+}$ or Mg-ATP binding. CorA/MRS2/ MGT channels constitute the major $Mg^{2+}$ uptake system in plants and are believed to be regulated similarly as their prokaryotic ancestors. Electrophysiological, electronic paramagnetic resonance

and molecular-dynamics studies of bacterial CorA show that cytoplasmic $Mg^{2+}$ acts as a ligand gating the pentameric channel (Dalmas et al., 2014). CorA crystal structures revealed that $Mg^{2+}$ binds to both the central pore and the intracellular region rich in acidic residues. When intracellular $Mg^{2+}$ is high (i.e., >5 mM), binding to sensor sites induces a closed conformation by increasing inter-subunit contacts. Conversely, when internal $Mg^{2+}$ drops and unbinding occurs, the channel undergoes asymmetric domain rearrangement, opening to allow $Mg^{2+}$ influx. In eukaryotic MRS2 (yeast or human), with resolved crystal structures, $Mg^{2+}$-sensing sites have been confirmed (Khan et al., 2013; Li et al., 2025). Plant homologs show structural similarity suggesting a comparable architecture, though direct functional gating studies (e.g., patch-clamp or conformational assays) are limited. Therefore, $Mg^{2+}$ regulation is likely, but experimental confirmation in plants is less extensive than in bacteria or human mitochondria.

Members of the CNNM/CorC family (including plant MGR proteins) carry cytosolic CBS-pair domains that in other systems bind Mg-ATP and trigger conformational changes that regulate transporter activity. Structural and functional work on bacterial/archaeal CorB/CorC and on animal CNNMs supports Mg-ATP as a regulatory ligand. High-resolution structures (apo and Mg-ATP-bound) of archaeal CorB/CorC reveal an ATP-binding site in the cytosolic domain and conformational differences between ligand-free and Mg-ATP-bound states. Binding induces a conformational rearrangement of the CBS dimer, which is communicated to the transmembrane DUF21 domain and alters the transporter's conductive or regulatory state (promoting or inhibiting Mg flux depending on the system). Functional assays indicate that Mg-ATP binding is important for $Mg^{2+}$ transport/export activity in these proteins (Chen et al., 2021). For plant MGRs, homology, domain architecture and physiological data strongly suggest similar regulation, but direct biochemical demonstration of Mg-ATP binding controlling plant MGR activity is absent in the peer-reviewed literature so far. MGRs have been mainly studied in Arabidopsis (9 members) and in wheat (15 members). In wheat, all 15 MGR genes contain conserved ABA-responsive elements in the promoter region (Chen et al., 2025) although direct regulation by ABA remains to be demonstrated. This observation is reminiscent of the clear overlap between Mg-deficiency-induced genes and ABA-responsive genes described by Hermans, Vuylsteke, Coppens, Cristescu, et al. (2010).

Additional regulation derives from proton-motive forces that drive antiporters such as MHX and from crosstalk with $Ca^{2+}$ signalling pathways that modulate transporter expression and ion channel activity. In Arabidopsis, tonoplast-localized $Ca^{2+}$ sensors, CBL2 and CBL3, contribute to $Mg^{2+}$ homeostasis via a V-ATPase-independent mechanism (Tang et al., 2015). CBL2 and CBL3 recruit a set of four functionally redundant CBL-interacting protein kinases (CIPK 3/9/23/26) to the tonoplast, establishing a CBL-CIPK signalling module that regulates vacuolar $Mg^{2+}$ sequestration in response to elevated cytosolic $Mg^{2+}$ (as seen in plants living on serpentine soils characterized by high $Mg^{2+}$ content and low $Ca^{2+}/Mg^{2+}$ ratios). The targets at the tonoplast of CIPK3/9/23/26 remain unidentified. Since the mutants of *MHX* or *MGT2/3* showed wild-type response to high $Mg^{2+}$, those transporters are not involved in this CBL-CIPK regulation. Because tolerance to high $Mg^{2+}$ was dependent on external $Ca^{2+}$ (Tang et al., 2015), Ródenas and Vert (2021) proposed that TPC1 (a slowly activated, non-selective, $Ca^{2+}$-activated vacuolar channel) is the target of CBL2/3-CIPK3/9/23/26. In strong support, the activity of TPC1 is modulated by high $Mg^{2+}$. In addition, the sucrose

non-fermenting-1-related protein kinase 2 (SnRK2) has recently been implicated in Arabidopsis high-$Mg^{2+}$ response (Mohamadi et al., 2023).

Genetic evidence reinforces the $Mg^{2+}$-$Ca^{2+}$ relationship: loss-of-function *cax1* mutants, lacking a major vacuolar $H^+/Ca^{2+}$ antiporter, exhibit enhanced tolerance to serpentine soils, likely because reduced vacuolar $Ca^{2+}$ sequestration increases cytosolic $Ca^{2+}$ available to counteract excess $Mg^{2+}$ (Bradshaw, 2005). Conversely, low external $Ca^{2+}$ supply can partially rescue $Mg^{2+}$-deficiency phenotypes in mutants lacking $Mg^{2+}$ transporters (Lenz et al., 2013).

Besides, $Mg^{2+}$ transporter activity can also be modulated by the ionic context of the environment. $Mg^{2+}$ flux inside chloroplasts illustrates how $Mg^{2+}$ transport operates within a broader ionic network (Figure 1). In illuminated chloroplasts, proton pumping into the thylakoid lumen during photosynthetic electron transport acidifies the lumen and generates a proton gradient between the stroma and the lumen. To maintain electrochemical balance, $Mg^{2+}$ and other cations (e.g., $K^+$) move from the lumen to the stroma, causing a rapid and reversible increase in free stromal $Mg^{2+}$. This process is pH-dependent, as light-induced stromal alkalinization is required for $Mg^{2+}$ release (Ishijima et al., 2003). Because several chloroplast enzymes, including Rubisco, depend on $Mg^{2+}$ for activation, this light-driven redistribution likely helps coordinate photosynthetic metabolism with light availability (Dukic et al., 2023). In addition, $Mg^{2+}$ flux affects the $\Delta$pH across the thylakoid membrane and contributes to the regulation of non-photochemical quenching (NPQ). $Mg^{2+}$ influx into the stroma may lower stromal pH through reversible cation/$H^+$ exchange across the chloroplast envelope (Dukic et al., 2023). KEA3-driven $H^+$ export from the thylakoid is also critical for rapid NPQ relaxation after a sudden decrease in light, and a potential $Mg^{2+}$ transport function of KEA3 would further link this process to stromal $Mg^{2+}$ dynamics (Uflewski et al., 2021).

A main strategy to study regulation of mineral homeostasis is to expose plants to nutrient deficiency. Understanding how plants sense and signal $Mg^{2+}$ deficiency remains challenging because stress signalling networks are inherently complex and often overlap with other physiological responses (Wilkins et al., 2016). Transcriptomic analyses under $Mg^{2+}$ starvation have identified numerous differentially expressed genes (DEGs), providing a global view of adaptive responses (Hermans, Vuylsteke, Coppens, Craciun, et al., 2010; Liang et al., 2024; Ogura et al., 2020; Yang et al., 2019). Although these DEGs do not directly reveal the molecular identity of $Mg^{2+}$ sensors, they highlight early-responding candidates potentially involved in deficiency signalling. $Mg^{2+}$ deficiency responses are spatially and temporally asynchronous, typically appearing first in roots before shoots (Hermans, Vuylsteke, Coppens, Cristescu, et al., 2010). Notably, $Ca^{2+}$ has emerged as a likely secondary messenger in $Mg^{2+}$ deficiency signalling. Several $Ca^{2+}$ transporter genes are upregulated during $Mg^{2+}$ starvation, and cytosolic $Ca^{2+}$ levels increase, consistent with the known $Mg^{2+}$-$Ca^{2+}$ antagonism (Hermans, Vuylsteke, Coppens, Craciun, et al., 2010; Hermans, Vuylsteke, Coppens, Cristescu, et al., 2010). Supporting this model, Wiesenberger et al. (2007) showed in yeast that $Mg^{2+}$ deficiency rapidly elevates cytosolic free $Ca^{2+}$, triggering the activation of $Ca^{2+}$-binding proteins (CaBPs), suggesting a central role for $Ca^{2+}$ in mediating $Mg^{2+}$-deficiency responses. Similar $Ca^{2+}$-dependent mechanisms appear to operate under $Mg^{2+}$ excess with the CBL-CIPK module (see above, Tang et al., 2015; Tang & Luan, 2017). Beyond indirect signalling effects, $Mg^{2+}$ directly modulates the activity of several ion channels, highlighting its dual role

as nutrient and signalling regulator. At the tonoplast, cytosolic $Mg^{2+}$ activates slow vacuolar (SV) channels while inhibiting fast vacuolar (FV) channels, reducing $K^+$ leakage and supporting ionic stability (Lemtiri-Chlieh et al., 2020). $Mg^{2+}$ also inhibits outward non-selective cation channels (NSCCs), such as MgC in guard and subsidiary cells of broad bean and maize, and participates in $NH_4^+/NH_3$ transport across the peribacteroid membrane in $N_2$-fixing plants, often coordinated with $Ca^{2+}$- signalling. At the plasma membrane, $Mg^{2+}$ further regulates $Ca^{2+}$ influx through hyperpolarization-activated $Ca^{2+}$ channels (HACCs), believed to correspond to cyclic nucleotide-gated channels (CNGCs). Physiological concentrations of cytosolic $Mg^{2+}$, mainly in the form of Mg–ATP, strongly inhibit HACC activity in guard cells at highly negative voltages ($\leq -200$ mV), via interaction with a conserved diacidic $Mg^{2+}$-binding motif (Lemtiri-Chlieh et al., 2020). Thus, $Mg^{2+}$ not only competes with $Ca^{2+}$ for permeation but also seems to fine-tune $Ca^{2+}$-dependent signalling.

Taken together, these findings reveal a tight interplay between $Mg^{2+}$ and $Ca^{2+}$ homeostasis, potentially involving shared transporters, regulatory sites and signalling components, although the underlying molecular mechanisms remain incompletely understood. Beyond its effects on $Mg^{2+}$-related pathways, $Ca^{2+}$ appears to act as a general mediator of nutrient signalling, influencing plant responses to $K^+$, $NO_3^-$, $B^{3+}$ and possibly $PO_4^{3-}$ (Behera et al., 2017; Matthus et al., 2019; Quiles-Pando et al., 2019; Wilkins et al., 2016; Xu et al., 2006).

Parallels can also be drawn with animal systems, where $Mg^{2+}$ functions as a second messenger regulating diverse cellular processes. For example, Li et al. (2011) showed that defective $Mg^{2+}$ flux underlies human T-cell immunodeficiency, while Stangherlin and O'Neill (2018) demonstrated that $Mg^{2+}$ dynamics modulate signal transduction. Collectively, these studies support the emerging view that $Mg^{2+}$, beyond its metabolic and structural roles, contributes to the fine-tuning of cellular signalling pathways in plants.

## 6. Interactions of magnesium with the circadian clock

Recent findings have revealed a dynamic role for $Mg^{2+}$ as a temporal regulator in plants. $Mg^{2+}$ appears to modulate the circadian clock, and reciprocally, the clock influences $Mg^{2+}$ homeostasis.

### 6.1. Overview of the plant circadian clock

Plants possess endogenous circadian clocks that generate ~24-hour rhythms, aligning internal processes with daily and seasonal environmental cycles. In Arabidopsis, up to 40% of the transcriptome exhibit rhythmic expression under constant conditions, highlighting the broad regulatory role of the clock (Covington et al., 2008; Rivière et al., 2024; Romanowski et al., 2020; Webb et al., 2019). These rhythms can be described by a sine wave function defined by three parameters: amplitude (the difference between the mean and peak height of the rhythm), phase (a fixed point in the cycle, relative to the external light and dark cycle) and period (the duration between two fixed points in the cycle, e.g., successive peaks). The core circadian oscillator is governed by clock genes forming interconnected transcriptional and post-translational feedback loops (TTFLs) driving downstream rhythmic growth and physiological outputs. The central oscillator is entrained by external cues (= zeitgebers) such as light and temperature, as well as internal signals, including ions and metabolites (Hsu & Harmer, 2014; Webb et al., 2019; Wang et al., 2022). Key components of the central oscillator include

morning-expressed genes *CIRCADIAN CLOCK-ASSOCIATED 1* (*CCA1*) and *LATE ELONGATED HYPOCOTYL* (*LHY*), morning-to-afternoon *PSEUDO-RESPONSE REGULATORS* (*PRRs*), *PRR9*, *PRR7* and *PRR5*, which are expressed sequentially throughout the day, and evening-expressed genes such as *TIMING OF CAB EXPRESSION 1* (*TOC1/PRR1*), the Evening Complex (*EARLY FLOWERING 3* or *ELF3*, *ELF4*, and *LUX ARRHYTHMO* or *LUX*), as well as *GIGANTEA* (*GI*) and *ZEITLUPE* (*ZTL*) (Greenwood & Locke, 2020; Hsu & Harmer, 2014; McClung, 2006).

Through these interconnected feedback loops, the circadian clock orchestrates key physiological processes, including stomatal opening and aquaporin activity, which regulate transpiration, the main driver of xylem nutrient transport, and photosynthesis, which produces sugars exported to the phloem to redistribute nutrients throughout the plant (Caldeira et al., 2014; Dodd et al., 2005; Haydon et al., 2015; Noordally et al., 2013; Westgeest et al., 2023). These rhythmic sugars, in turn, feedback to entrain the clock, primarily via PRR7, a strong inhibitor of CCA1, which is repressed by photosynthetically derived sugars. Under low-energy conditions, exogenous sucrose shortens the circadian period in a PRR7-dependent manner, maintains oscillations in continuous darkness, and advances or delays clock phase depending on the timing of sugar application, whereas *prr7* mutants are largely insensitive to these effects (Knight et al., 2008; Haydon et al., 2013; Liu et al., 2013; Frank et al., 2018; Wang et al., 2022). Notably, approximately 40% of sugar-responsive genes exhibit circadian rhythmicity (Bläsing et al., 2005).

## 6.2. Circadian regulation of ion homeostasis

Historically, circadian rhythms were hypothesized to arise from feedback loops involving ion gradients and membrane transporters (Nitabach et al., 2005; Njus et al., 1974). While the gene-centric model now dominates in circadian research, ion fluxes remain a critical interacting layer, that both modulate and is modulated by transcriptional and non-transcriptional feedback loops (Henslee et al., 2017; Mihut et al., 2025). Mineral nutrients are now attracting renewed interest for their ability to influence circadian timing and improve crop productivity through chronoculture (Hastings et al., 2008; Ogasawara et al., 2025; Steed et al., 2021).

Nutrient availability can affect clock dynamics in diverse ways (Table 1). Nitrogen deficiency shortens the circadian period in the dinoflagellate *Gonyaulax polyedra* (Sweeney & Folli, 1984), and $NO_3^-$ pulses entrain circadian gene expression in Arabidopsis, where clock components regulate many nitrogen-responsive genes (Covington et al., 2008; Gutiérrez et al., 2008; Haydon et al., 2011; Porco et al., 2024). Transcripts associated with transport and homeostasis of $K^+$, $PO_4^{3-}$ and $SO_4^{2-}$ are also circadian regulated (Lebaudy et al., 2008; Haydon et al., 2011; Wang et al., 2011; Cao et al., 2013; Uemoto et al., 2023). In Arabidopsis, cytosolic and chloroplastic $Ca^{2+}$ display circadian oscillations, peaking between midday and dusk in mesophyll cells (Johnson et al., 1995; Love et al., 2004; Martí Ruiz et al., 2020; Sai & Johnson, 2002). Although some $Ca^{2+}$ channels and transporters exhibit rhythmic transcription, these oscillations are thought to be primarily driven by post-transcriptional regulation (Dodd et al., 2007; Haydon et al., 2011, 2015).

Among micronutrients, $Cu^{2+}$ and $Fe^{2+}$ transport and homeostasis are under circadian regulation (Duc et al., 2009; Hong et al., 2013; Perea-García et al., 2010). Transcripts for $Cu^{2+}$ transporters show circadian rhythms and their promoters contain conserved circadian elements (Covington et al., 2008; Dodd et al., 2007;

Perea-García et al., 2010). Excess $Cu^{2+}$ reduces clock amplitude and may lengthen circadian period, likely through GI-dependent pathways (Andrés-ColÁs et al., 2010; Haydon et al., 2011).

Similarly, genes involved in $Fe^{2+}$ transport and storage, including ferritin-encoding genes are regulated by both $Fe^{2+}$ availability and circadian components such as PRR7 and TIME FOR COFFEE (TIC) (Ding et al., 2007; Duc et al., 2009; Liu et al., 2013). $Fe^{2+}$ deficiency lengthens the circadian period by ~1–2 hours in a light-dependent manner, requiring evening components such as GI and ZTL, while morning components CCA1 and LHY directly regulate $Fe^{2+}$ uptake genes (Chen et al., 2013; Hong et al., 2013; Salomé et al., 2013; Xu et al., 2019).

### 6.2.1. Magnesium oscillations and clock function. Magnesium is currently the nutrient with the most well-characterized connection to the circadian clock (Siqueira et al., 2023). Oscillations in intracellular $Mg^{2+}$ levels have been observed across multiple organisms, including the green unicellular alga *Ostreococcus tauri*, the fungus *Neurospora crassa* and human U2OS cells (Feeney et al., 2016), as well as at the subcellular level in rice chloroplasts (Li et al., 2020; Chen et al., 2022). These oscillations influence global translational activity by modulating ATP stability and ribosomal function (de Barros Dantas et al., 2023; Feeney et al., 2016). $Mg^{2+}$ rhythms peak around dusk in Ostreococcus but at dawn in rice chloroplasts, suggesting that $Mg^{2+}$ regulation differs among species and subcellular compartments (Feeney et al., 2016; Li et al., 2020).

In cyanobacteria, $Mg^{2+}$ directly modulates the KaiC-based oscillator, and artificial $Mg^{2+}$ cycles can drive rhythmic gene expression even in the absence of core clock genes *KaiA* and *KaiB*, highlighting the importance of non-transcriptional circadian regulation (Rust et al., 2011; Jeong et al., 2019; Li et al., 2022). Similarly, in Ostreococcus, intracellular $Mg^{2+}$ rhythms persist without transcriptional activity in constant darkness, supporting post-translational control of $Mg^{2+}$ transport and $Mg^{2+}$ oscillations (Feeney et al., 2016). These observations emphasize the importance of non-gene-centric mechanisms in circadian regulation.

In mammals, the $Mg^{2+}$ transporter TRANSIENT RECEPTOR POTENTIAL CATION CHANNEL SUBFAMILY M MEMBER 7 (TRPM7) exhibits rhythmic expression that affects intracellular $Mg^{2+}$ levels, linking $Mg^{2+}$ homeostasis to the mammalian circadian clock (Uetani et al., 2017; Zhang et al., 2021).

In Arabidopsis, $Mg^{2+}$ deficiency consistently lengthens the circadian period and dampens oscillation amplitude, a response also conserved in Ostreococcus and human U2OS cells (de Melo et al., 2021; Feeney et al., 2016; Hermans, Vuylsteke, Coppens, Craciun, et al., 2010). This effect is amplified under long-day photoperiods and in the presence of sucrose, suggesting that both light and metabolic status modulate the interaction between $Mg^{2+}$ and the clock (de Melo et al., 2021; Hermans, Vuylsteke, Coppens, Cristescu, et al., 2010). $Mg^{2+}$ also interacts with light-perception pathways, notably phytochromes, feeding back to regulate the clock (Rivière et al., 2021). However, unlike $NO_3^-$, $Mg^{2+}$ pulses fail to reset circadian phase, suggesting that $Mg^{2+}$ does not act as an entrainment signal in Arabidopsis (de Melo et al., 2021).

In rice, chloroplast $Mg^{2+}$ oscillations are directly circadian-regulated via *OsMGT3* (AtMGT10), which exhibits rhythmic expression peaking around dawn (J. Li et al., 2020), and is repressed by two PRR proteins (OsPRR59, OsPRR95) (Chen et al., 2022). *OsMGT3* rhythmic expression generates diel fluctuations of $Mg^{2+}$ in rice chloroplasts, influencing Rubisco activity and photosynthetic carbon fixation rates (C.-Q. Chen et al., 2022). Knockout mutants *osmgt3* show reduced chloroplast $Mg^{2+}$ rhythms and

**Table 1.** Interactions between nutrients and the plant circadian clock. Information is for Arabidopsis unless specified.

| Nutrient | Nutrient effect on clock (clock parameters) | Clock effect on nutrient (nutrient levels, transport or homeostasis) | Reference |
|---|---|---|---|
| Calcium | • Deficiency may lengthen the period | • Oscillation of calcium levels in cytosol and chloroplast<br>• Oscillation of several transporters/regulators | Johnson et al., 1995; Love et al., 2004; Xu et al., 2007; Haydon et al., 2011; Martí Ruiz et al., 2020; Uemoto et al., 2023 |
| Magnesium | • Deficiency lengthens the period<br>• Deficiency dampens the amplitude<br>• Pulse does not shift the phase in Arabidopsis but in Ostreococcus) | • Oscillation of magnesium levels in chloroplast | Hermans, Vuylsteke, Coppens, Craciun, et al., 2010; Feeney et al., 2016; de Melo et al., 2021; Hargreaves et al., 2022 |
| Nitrogen | • Deficiency shortens the period in Gonyaulax but not in Arabidopsis<br>• Deficiency shortens the phase<br>• Pulse shift the phase | • Oscillation of several transporters/regulators | Sweeney & Folli, 1984; Tucker et al., 2004; Gutiérrez et al., 2008; Teng et al., 2017; de Melo et al., 2021; Chen et al., 2022; Yang et al., 2022; Liu et al., 2024; Porco et al., 2024 |
| Potassium | • Coordinates circadian rhythms of leaf growth<br>• Deficiency may lengthen the period | • Oscillation of several transporters/regulators | Kondo et al., 1983; Haydon et al., 2011; Lu et al., 2022; Uemoto et al., 2023 |
| Phosphorus; sulphur | | • Oscillation of several transporters/regulators | Harmer et al., 2000; Haydon et al., 2011; Moshelion et al., 2002; Versaw et al., 2002; Cao et al., 2013 |
| Copper | • Deficiency may lengthen the period<br>• Deficiency increases the amplitude | • Oscillation of several transporters/regulators | Andrés-Colás et al., 2010; Perea-García et al., 2010; Haydon et al., 2011 |
| Iron | • Deficiency lengthens the period<br>• Pulse shift the phase | • Oscillation of several transporters/regulators | Ding et al., 2007; Duc et al., 2009; Chen et al., 2013; Hong et al., 2013; Liu et al., 2013; Salomé et al., 2013; Xu et al., 2018 |

photosynthetic efficiency, while mesophyll-specific overexpression enhances growth and carbon assimilation (Li et al., 2022). This remains the only confirmed example of a clock-controlled $Mg^{2+}$ transporter in plants.

### 6.3. Magnesium, metabolism and circadian integration

Magnesium's regulation of circadian rhythms is deeply intertwined with its role in cellular energy metabolism. As the physiological cofactor of ATP, $Mg^{2+}$ is required for most energy-dependent reactions, including photosynthesis, ATP synthase activity and translation (Kleczkowski & Igamberdiev, 2021; Ko et al., 1999). In chloroplasts, ATP synthesis increases with external $Mg^{2+}$ supply in a light-dependent manner (Lin & Nobel, 1971; Ishijima et al., 2003; Chen et al., 2018), whereas $Mg^{2+}$ deficiency causes ADP accumulation, impaired respiration and growth arrest (Gout et al., 2014).

Through its tight coupling with adenylate metabolism, $Mg^{2+}$ functions upstream of the Target of Rapamycin (TOR) pathway, a conserved kinase energy-sensor complex that modulates the circadian clock and integrates nutrient, hormone and environmental signals to promote growth and translation while repressing autophagy in both mammals and plants (Khapre et al., 2014; Feeney et al., 2016; Wu et al., 2019; Zhang, Meng, et al., 2019; Liu et al., 2021; Meng, Zhang, Li, et al., 2022; Urrea-Castellanos et al., 2025). Mammalian TOR activity is highly $Mg^{2+}$-dependent, since MgATP functions as its substrate and an additional $Mg^{2+}$ ion binds the active site to enable target phosphorylation (Feeney et al., 2016; Kleczkowski & Igamberdiev, 2023). TOR form an antagonistic regulatory module with the Sucrose non-fermenting-Related protein Kinase 1 (SnRK1). TOR inhibits SnRK1 signalling under energy-replete conditions, whereas SnRK1 suppress TOR activity during energy-deplete conditions, shifting metabolism from growth towards conservation, autophagy and stress responses. Activated SnRK1 phosphorylates and activates the transcription factor basic leucine Zipper 63 (bZIP63), which in turn binds to the *PRR7* promoter to increase its transcription. Elevated PRR7 expression delays circadian phase and is required for sugar-induced shortening of the circadian period (Figure 2) (Haydon et al., 2013; Frank et al., 2018).

In human and algal cells, TOR inhibition lengthens the circadian period and abolishes the $Mg^{2+}$-depletion lengthening effect, indicating that $Mg^{2+}$ regulates the clock, at least in part, via TOR signalling (Feeney et al., 2016; Rubin, 2007; van Ooijen & O'Neill, 2016). In Arabidopsis, TOR inhibition similarly lengthens the circadian period in a dose-dependent manner (Zhang, Meng, et al., 2019; Wang, Qin, et al., 2020; Urrea-Castellanos et al., 2022). Restoration of sugar under low-energy condition reactivates TOR and normalizes/shortens period length, whereas this recovery is lost when TOR is silenced. Conversely, nicotinamide, a precursor of $NAD^+$, inhibits sugar-driven ATP production, suppresses TOR activity, and lengthens the circadian period. Together, these observations demonstrate that both sugar-induced period shortening and nicotinamide-induced period lengthening are TOR-dependent (Figure 2) (Dodd et al., 2007; Haydon et al., 2013; Zhang, Meng, et al., 2019; de Melo et al., 2021). $Mg^{2+}$ deficiency also delays the phase of PRR7 expression, which may relate to the finding that *prr7-11* mutation abolishes nicotinamide- and sugar-induced clock adjustments (de Melo et al., 2021; Farré & Weise, 2012; Haydon et al., 2013; Mombaerts et al., 2019). The convergent period-lengthening effects of $Mg^{2+}$ deficiency, sugar deprivation and nicotinamide, along with their shared

TOR-dependent influence on circadian timing, indicate that $Mg^{2+}$ most likely modulates the circadian clock, at least in part, upstream of TOR signaling through conserved regulatory pathways in plants (Figure 2).

Conversely, there is growing evidence that the circadian clock exerts feedback on TOR signalling. In *bzip63* mutants, rhythmic expression of *Ribosomal Protein S6 Kinase 1 (S6K1)* – a key downstream effector of TOR that phosphorylates ribosomal protein S6 to stimulate translation and cell growth – is disrupted, indicating that TOR signalling is at least partly under circadian control (Frank et al., 2018; Urrea-Castellanos et al., 2022). Moreover, PRR proteins (specifically PRR5, PRR7 and PRR9) repress the transcription of *TANDEM ZINC FINGER 1 (TZF1)*; this repression prevents TZF1-mediated destabilization of TOR mRNA, thereby maintaining TOR signalling (Li et al., 2019; Wang, Qin, et al., 2020).

Sucrose influences the plant circadian clock in a context-dependent manner. Under low-energy conditions, sucrose shortens the circadian period, but has little impact in energy-repleted condition. Likewise, TOR signalling is activated by sugar availability but is repressed during sugar starvation or sugar excess, reflecting the complex integration of cellular energy status, sugar specificity, and stress signals in plants. Together, these observations highlight the intricate interplay between ionic cues, metabolic cues and circadian regulation (Haydon et al., 2013; Han et al., 2022; Wang et al., 2022; Pereyra et al., 2023). Therefore, the stronger impact of $Mg^{2+}$ deficiency under sucrose-supplemented and long-day conditions (de Melo et al., 2021) likely arises from interactions between nutrient limitation and cellular energy status. Importantly, $Mg^{2+}$ also plays a critical role in sucrose transport, further linking $Mg^{2+}$ availability to carbon metabolism.

$Mg^{2+}$ status, metabolism and circadian regulation also influence stomatal movement. TOR negatively regulates abscisic acid (ABA) signalling by phosphorylating Pyrabactin Resistance 1/PYR1-Like/Regulatory Component of ABA Receptor (PYR/PYL/RCAR) receptors, favouring growth over stress responses. Upon stress, ABA accumulates and binds to PYR/PYL receptors, inhibiting type 2C protein phosphatases (PP2Cs), and activating SnRK1 and SnRK2 kinases, which phosphorylate downstream targets and inhibit TOR (Figure 2) (Baena-González & Hanson, 2017; Belda-Palazón et al., 2020, 2022).

$Mg^{2+}$ deficiency upregulates numerous ABA-responsive genes (Hermans et al., 2004; Hermans, Vuylsteke, Coppens, Cristescu, et al., 2010), whereas $Mg^{2+}$ excess activates a subclass of SnRK2 kinases in an ABA-dependent manner which, together with $Ca^{2+}$-binding proteins, contributes to the maintenance of $Mg^{2+}$ homeostasis, likely via vacuolar transport (Mogami et al., 2015).

Upon ABA perception, activated SnRK2s activate slow anion channel-associated 1 (SLAC1) and related ion channels, promoting anion efflux and membrane depolarization, a key step in stomatal closure (Wang et al., 2018; Wei et al., 2025). ABA also triggers cyclic ADP-ribose (cADPR)-mediated $Ca^{2+}$ release, generating transient cytosolic $Ca^{2+}$ spikes that reinforce stomatal closure (Wu et al., 1997; Leckie et al., 1998). Notably, stomatal movement is known to be modulated by sugar, nicotinamide and TOR activity. TOR inhibition impairs light-induced stomatal opening, while nicotinamide, an inhibitor of both cADPR synthesis and TOR activity, inhibits ABA-induced stomatal closure in a dose-dependent manner (Han et al., 2022; Kottapalli et al., 2018; Leckie et al., 1998; Siegel et al., 2009). In addition, cytosolic $Mg^{2+}$ levels, regulated in part by the tonoplast-localized $Mg^{2+}$ transporter MGR1, are essential for stomatal opening, with vacuolar $Mg^{2+}$ sequestration particularly important under high $Mg^{2+}$ conditions (Inoue et al., 2022).

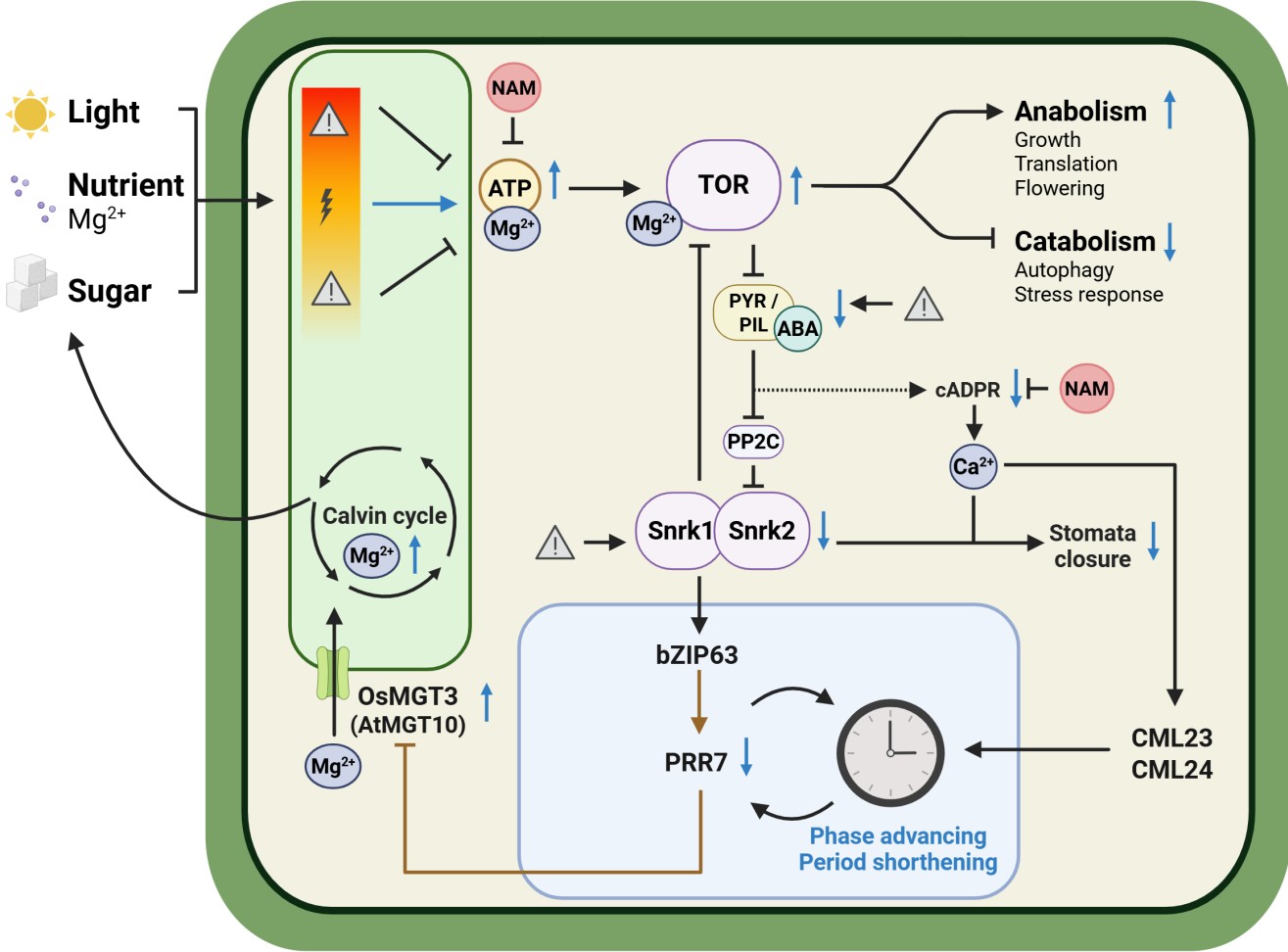

**Figure 2.** Hypothetical working model linking magnesium homeostasis, metabolism, calcium signaling, stomatal regulation and circadian clock in plants.

Under optimal conditions of light, sugar and nutrient availability, TOR is activated. TOR promotes anabolic processes while repressing catabolic pathways, partially through inhibition of abscisic acid (ABA) signalling. TOR-mediated suppression of ABA signalling inhibits SnRK1 and SnRK2 kinases, which modulate the bZIP63-PRR7 regulatory module. This module feeds back to the circadian clock, resulting in phase advancement and period shortening.

TOR also represses SnRK2 and cADPR-dependent $Ca^{2+}$ signalling pathways, promoting stomatal opening under favourable growth conditions. Oscillations in $Ca^{2+}$ are sensed by $Ca^{2+}$-binding proteins CML23 and CML24, which in turn feed back to the circadian clock.

In rice, the rhythmically expressed $Mg^{2+}$ transporter gene *OsMGT3* (*AtMGT10* in Arabidopsis) is transcriptionally repressed by PRR proteins; disruption of this regulation diminishes $Mg^{2+}$ oscillations, photosynthesis and overall growth.

Arrow codes: Black solid arrows, established regulation; brown solid arrows, established transcriptional regulation; black dashed arrows, proposed or indirect regulation requiring further validation; blue arrows, upregulation or downregulation under optimal growth conditions. Lightning bolt = optimal growth conditions, warning sign = stress conditions (e.g., energy or nutrient limitation) and NAM = nicotinamide *(figure created with BioRender.com )*.

Cytosolic and stromal $Ca^{2+}$ levels display circadian rhythmicity, peaking around dusk in mesophyll cells, and feed back to the clock via Calmodulin-Like proteins 23 and 24 (CML23 and CML24), which are upregulated by ABA and darkness (Johnson et al., 1995; Love et al., 2004; Delk et al., 2005; Ruiz et al., 2018; Frank et al., 2019). These oscillations are thought to depend on cADPR-mediated $Ca^{2+}$ release from internal stores, as in animals, although the plant cADPR-sensitive channel and cADPR cyclase remain unidentified (Dodd et al., 2007; Ikeda et al., 2003). Strikingly, sucrose starvation and nicotinamide, which inhibit $Mg^{2+}$-sensitive TOR signalling, not only lengthen the circadian period and alter stomatal movement but are also known to abolish $Ca^{2+}$ oscillations (Figure 2) (Dodd et al., 2007; Johnson et al., 1995; Martí Ruiz et al., 2020).

Together, these findings support a model in which $Mg^{2+}$, $Ca^{2+}$, $NAD^+$, sugars and TOR form an integrated metabolic–ionic network that couples cellular energy status to circadian regulation

(Figure 2). $Mg^{2+}$ acts as a metabolic integrator, connecting adenylate pools to TOR signalling and circadian timing. Daily fluctuations in nutrients and metabolites, including $Mg^{2+}$, $Ca^{2+}$ and sugars, provide a direct mechanism by which changes in cellular energy are mirrored into circadian adjustments, feeding back to modulate clock function, stomatal dynamics, and stress responses. This framework highlights a complex non-transcriptional layer of circadian regulation, in which ionic and metabolic cues converge to coordinate temporal control of growth, metabolism, and environmental responses.

## 7. Conclusion and perspectives

$Mg^{2+}$ plays a pivotal role in plant physiology, acting both as a structural element and as a versatile regulatory cofactor. It is essential for chlorophyll coordination, activation of ATP-dependent enzymes and maintenance of ion homeostasis and signalling processes.

Beyond these classical roles, recent findings have revealed that $Mg^{2+}$ also influences circadian regulation, positioning it as an integrator of temporal, metabolic and nutritional cues in plants.

Despite its broad functional spectrum, $Mg^{2+}$ remains comparatively understudied relative to other macronutrients. Considerable progress has been made in identifying $Mg^{2+}$ transporters, yet major questions remain regarding their regulation, subcellular dynamics and integration into whole-plant nutrient allocation strategies. The complex interactions of $Mg^{2+}$ with other ions, such as $K^+$, $Ca^{2+}$, $NH_4^+$ and transition metals, represent another underexplored frontier, both at physiological and molecular scales.

In agronomic contexts, $Mg^{2+}$ deficiency remains a widespread and often overlooked issue, particularly in acidic or intensively cultivated soils. Improving crop resilience and nutrient use efficiency will require a deeper understanding of how $Mg^{2+}$ uptake, redistribution and sensing are modulated under stress and throughout plant development. Such knowledge will be instrumental for optimizing fertilization practices and guiding breeding strategies towards genotypes with enhanced Mg efficiency.

The recently established link between $Mg^{2+}$ and circadian regulation opens promising research directions. $Mg^{2+}$ may contribute to adaptive rhythmicity, fine-tuning metabolic and physiological processes in response to nutrient availability and environmental fluctuations. Investigating how $Mg^{2+}$ homeostasis interacts with diurnal cycles and metabolic rhythms could uncover new regulatory mechanisms relevant to growth, yield and stress adaptation.

Future research should focus on the post-transcriptional and post-translational regulation of $Mg^{2+}$ transporters, including the identification of upstream regulators such as kinases, phosphatases, small RNAs and protein interactors. Understanding how these components integrate into broader nutrient signalling and environmental response networks, such as those governed by circadian cues, light quality, or abiotic stress, will be key to grasping the dynamic control of $Mg^{2+}$ homeostasis. Finally, exploring natural variation in transporter regulation across species or genotypes adapted to contrasting soils may reveal evolutionary strategies for $Mg^{2+}$ acquisition and utilization, providing valuable targets for sustainable crop improvement.

**Open peer review.** To view the open peer review materials for this article, please visit http://doi.org/10.1017/qpb.2025.10036.

## Acknowledgements

The authors thank Professor Alex Webb (University of Cambridge) and Dr. Quentin Rivière (Biology Centre of the Czech Academy of Sciences, Institute of Plant Molecular Biology) for their insightful feedback on the manuscript and the two anonymous reviewers for their constructive comments.

**Competing interests.** The authors declare none.

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
