## [Reviewer Report]

This is an excellent, timely and well-written review that encapsulates many aspects of magnesium as an essential ion in plants. It is particularly useful to remind the reader that Mg accumulation in plants is critical to the human diet and human health, as outlined in the Introduction. Equally, coverage of literature on the role of Mg in control of the circadian clock is a useful and insightful last section.

I believe that the review could be published as it stands. However, the review would be considerably strengthened were more quantitative information given on homeostasis of Mg in the cytosol and the chloroplast stroma. These are two contrasting cases. In the cytosol, it is essential that the free Mg levels are tightly controlled around set points to ensure that catabolic reactions are provided with a sustained energy charge – circadian oscillators notwithstanding. A small sub-section on this aspect that would expand on what is known about Mg transporters and feedback systems would help: how is free Mg tightly regulated in the cytosol? In the contrasting case - the stroma - Mg release from thylakoids has long been recognised as the trigger to initiate carbon assimilation in the light. Again, some numbers would be useful relating to steady-state stromal Mg in the light and dark and the driving forces that trigger thylakoidal Mg release in the light. In this context, it would be useful to expand Section 2, particularly highlighting gaps in our knowledge around the control of free Mg levels in the cytosol.

Otherwise, this MS is well-suited for publication in QPB.

---

## [Reviewer Report]

This is a very well written and very exhaustive review on the role of magnesium in plants with a special emphasis on its contribution to circadian regulation. There is not much to improve besides a few typos that might be corrected:

*) p.1 “the most abundant divalent cations” -> “the most abundant divalent cation”

*) p.18 “the dynamic role of as Mg²⁺ as” -> “the dynamic role of Mg²⁺ as”

*) p.20 “where clock components regulates many” -> “where clock components regulate many”

*) p.23 “Mg²⁺ pulses does not reset the clock” -> “Mg²⁺ pulses do not reset the clock”

---

## [Editor Report]

Dear Charlotte and Nathalie, 

your manuscript “ Magnesium: An Overlooked Signaling Ion in Plant Physiology and Circadian Regulation” has been seen by two independent reviewers. Both are very positive and have only minor comments. One of the reviewers has a few suggestions for improvement, for instance to add some more quantitative information on homeostasis of Mg in the cytosol and the chloroplast stroma. Please check whether you may handle these suggestions in a minor revision. Thank you for your valuable contribution to the Research Topic “Quantitative approaches to cellular aspects of plant ion homeostasis”.

Best wishes

Ingo

---

## [Editor Report]

Dear Charlotte and Nathalie,

thank you for the careful revision of the manuscript and thanks again for your valuable contribution to the Research Topic “Quantitative approaches to cellular aspects of plant ion homeostasis”. It is highly appreciated. Please submit the graphical abstract; it appears blank in all files.

Best regards, Ingo